# Features of the Composition and Photoluminescent Properties of Porous Silicon Depending on Its Porosity Index

**Aleksandr S. Lenshin** [1,2,*], **Yaroslav A. Peshkov** [1], **Konstantin A. Barkov** [1], **Margarita V. Grechkina** [1], **Anatoliy N. Lukin** [1], **Sergey V. Kannykin** [1], **Dmitriy A. Minakov** [1] and **Olga V. Chernousova** [2]

[1] Physical Department, Voronezh State University, 394000 Voronezh, Russia
[2] Chemical Department, Voronezh State University of Engineering Technologies, 394000 Voronezh, Russia
[*] Correspondence: lenshinas@mail.ru

**Abstract:** Porous silicon samples with a porosity index of 5% to 80% were obtained in this work by electrochemical etching, and their photoluminescence properties were also studied. The porosity index was calculated according to the data from X-ray reflectometry. The composition of the surface was controlled by ultra-soft X-ray spectroscopy and infrared (IR) spectroscopy. The degree of the sample surface oxidation increased with the porosity enhancement. Two known mechanisms of photoluminescence in porous silicon were detected that are related to the composition and morphology of its surface. The values of the porosity index specifying the dominations of these mechanisms were determined. Enhancement of photoluminescence was shown to be attributed to an increase in the porosity index.

**Keywords:** porous silicon; photoluminescence; X-ray reflectometry; porosity; oxidation; spectroscopy

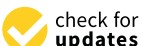



## 1. Introduction

Porous silicon (por-Si) is a compound multiphase material and its composition and functional properties are highly dependent on the conditions of its formation [1–3]. Porous silicon is composed of crystalline and amorphous silicon, as well as silicon oxides with different oxidation degrees ($SiO_x$), a dependence on the technology of its fabrication. The surface of porous silicon also involves Si-OH bonds as well as silicon–hydrogen bonds [2,4]. The most interesting properties of por-Si in terms of its practical use in optoelectronics, sensor applications, and biomedicine that distinguish it from the bulk crystalline silicon are its high specific surface area, characterized by porosity index and pore sizes, as well as photoluminescence (PL) in the visual spectral range. The quantum-size effect is considered a main PL mechanism [4]. It is observed as a bright luminescence with a peak in the range of 600–700 nm. Moreover, the luminescence properties of por-Si can be detected in the range of 500–600 nm, which are due to the radiative recombination centers formed in the defect oxide covering the surface of porous silicon. Usually, it is a broad luminescence band with relatively low intensity [5].

In our work, we studied the composition and photoluminescence properties of porous silicon samples with different porosity indexes. X-ray reflectometry (XRR) is of a certain significance among the analytical techniques for the calculation of the por-Si porosity index [6]. The advantage of XRR utilization is that it allows for the determination of the porosity index in the surface layer, thereby determining the photoluminescence properties of por-Si. The porosity index was shown to be connected with the depth of analysis and the time of a sample exposure in the atmosphere, as shown in [7,8]. Our work aimed to find correlations between the porosity index, composition, intensity and mechanism of photoluminescence in porous silicon. Therefore, we attempted to select the analytical techniques for the determination of the porosity index and composition of the layer where just photoluminescence is excited.

## 2. Materials and Methods

Porous silicon samples were obtained by electrochemical etching (ECE) of single-crystalline silicon plates of n- (KEF) and p-type (KDB) in a solution of hydrofluoric acid, isopropyl alcohol and hydrogen peroxide according to the methods described in [9]. The parameters of the etching procedures are presented in Table 1. Varying of samples porosity was employed by the step-wise change of the current density during chemical anodic treatment in the ECE process, and the choice of the original single-crystalline plate; the composition of the etching solution remained invariable. It is well-known that degradation processes connected to the changes in composition and properties of the porous layer proceed very rapidly in the early days after its fabrication. Therefore, the main investigations were performed in the several months after obtaining the samples when their composition and photoluminescence (PL) were stabilized [4].

**Table 1.** Fabrication parameters for obtaining porous silicon samples and the results of the study of porosity and photoluminescence.

| № of a Sample | Type and Resistivity of Original Si Plate, Ω·m | Current Density for ECE, mA | ECE Time, min | Porosity, % | Relative Intensity of PL Peak, % |
|---|---|---|---|---|---|
| 258 | KDB, 0.5 | 25 | 9 | 5 | 3 |
| 259 | KDB, 5 | 25 | 9 | 14 | 4 |
| 253 | KEF, 0.2 | 50 | 9 | 32 | 85 |
| 254 | KEF, 0.2 | 50/20/20 | 3/3/3 | 54 | 94 |
| 255 | KEF, 0.2 | 20/50/20 | 3/3/3 | 80 | 100 |

AFM images were obtained with a SOLVER P47 NT-MDT microscope. They were performed 3 months after obtaining the samples.

To measure the values of porosity, X-ray reflectometry was employed for the samples of por-Si with the use of an ARL X'TRA X-ray diffractometer operating in a Bragg–Brentano scheme (with Cu K$\alpha$ radiation). An error in the reflections' position relative to the standard one did not exceed 0.010° (2$\theta$). The position of the critical angle of the total external reflection (TRA) is proportional to the mean value of the electron density of the medium [10]. Therefore, the value of the critical angle of TRA for porous silicon $\theta_{c\text{-}PS}$ and silicon substrate $\theta_{c\text{-}Si}$ makes it possible to calculate the porosity index by the ratio of $P(\%) = [1 - (\theta_{c\text{-}PS}/\theta_{c\text{-}Si})^2] \cdot 100$ [11]. The penetration depth of the X-rays into por-Si close to the critical angle is about several tenths of nanometers, which is comparable with the depth of PL excitation [2,4].

The photoluminescence properties were studied with the use of fiber-optic spectrometer Ocean Optics USB 4000-VIS-NIR under excitation with a source operating at the wavelength of 405 nm.

The ultra-soft X-ray emission spectroscopy USXES technique widely employed for the study of the electron structure in disordered systems is very useful for the study of the composition of systems of this kind since it provides information on the nearest chemical environment of the atoms in a material. It allows for the chemical bond characteristics of the phases presented above as well as their ratio at a depth of analysis from ~10 to ~120 nm [12].

The Si L$_{2,3}$-spectra of the investigated porous silicon samples were obtained with a RSM-500 X-ray spectrometer-monochromator allowing us to study the spectra within the wavelength range of 0.5–50 nm. The depth of analysis for the samples was 60 nm at the energy of electrons exciting characteristic X-ray radiation equal to 3 keV [12]. Simulation of USXES spectra was performed with the use of weight factors employing our original program. While simulating the Si L$_{2,3}$-spectra of por-Si samples, we used the reference spectra of single-crystalline silicon c-Si, amorphous hydrogenated silicon a-Si:H, silicon suboxide SiO$_x$ ($x$~1,3) and silicon dioxide SiO$_2$ [12,13]. An error of simulation was determined as the difference between the area under experimental and simulated Si L$_{2,3}$-spectrum and

it did not exceed 10%. The investigations were performed half a year after the fabrication of the samples.

To obtain data on the chemical bonds and their possible deformations on the surface of por-Si samples, additional studies were performed with the use of infrared (IR) spectroscopy technique. IR transmission spectra of porous silicon samples were obtained with a Vertex 70 (Bruker) IR Fourier spectrometer utilizing the ATR attachment.

## 3. Results and Discussion

To determine the surface porosity of por-Si samples X-ray reflectometry technique was applied. The XRR profiles for five samples are presented in Figure 1. One can see that each por-Si is characterized by its own critical angle of the total external reflection $\theta_{c\text{-}PSi}$. Note that all of the critical angles for the samples are below the critical angle of the crystalline silicon ($\theta_{c\text{-}Si} \approx 0.22°$ for $\lambda = 1.54$ Å)

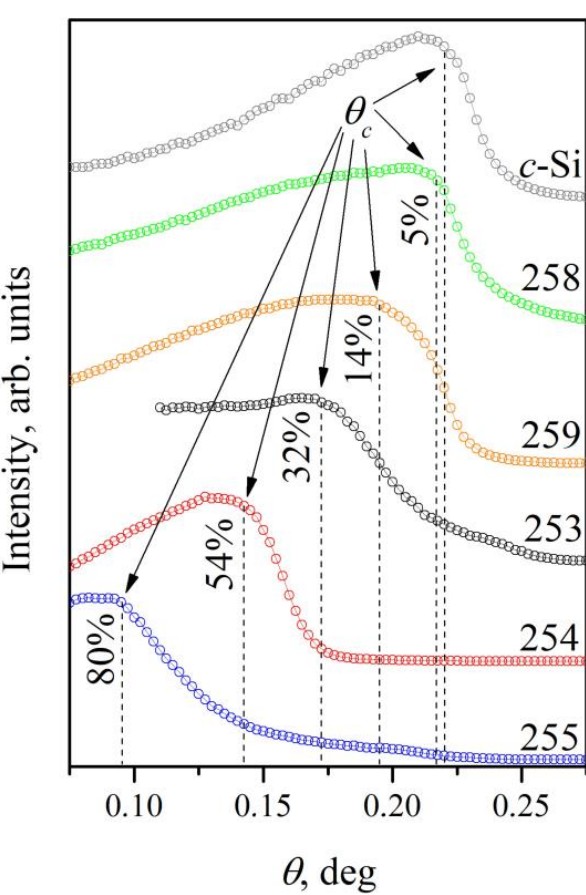

**Figure 1.** XRR curves and the calculated porosity indexes for the samples of porous silicon. The dashed lines show the values of critical angles corresponding to the total external reflection $\theta_{c\text{-}PS}$.

Calculations of the porosity values for the samples show that by a selection of original crystalline silicon substrate and a suitable mode of electrochemical etching, it is possible to successfully control the value of porosity factor for por-Si in the range of 5% to 80% (Table 1).

SEM images of the cleavages of porous silicon samples obtained at the different modes of electrochemical etching of crystalline silicon substrates of KEF (n-) and KDB (p-) type with an <100> orientation are presented in Figure 2 (see Table 1). The data on the morphology of the samples are in a very good visual agreement with the XRR data. Samples with a greater porosity index looked as if they were more disordered; the mean size of the surface inhomogeneity increased (from ~50 to 100 nm). Depending on the choice of the original c-Si plate and ECE mode, the thickness of porous layers in the samples was

~10 µm. As our task was to obtain samples with different porosity indexes at the sample surface and study of interrelations of the surface characteristics in the structures (up to 100 nm in depth), such a spread in the thickness over investigated samples did not provide any additional errors when considering our conclusions.

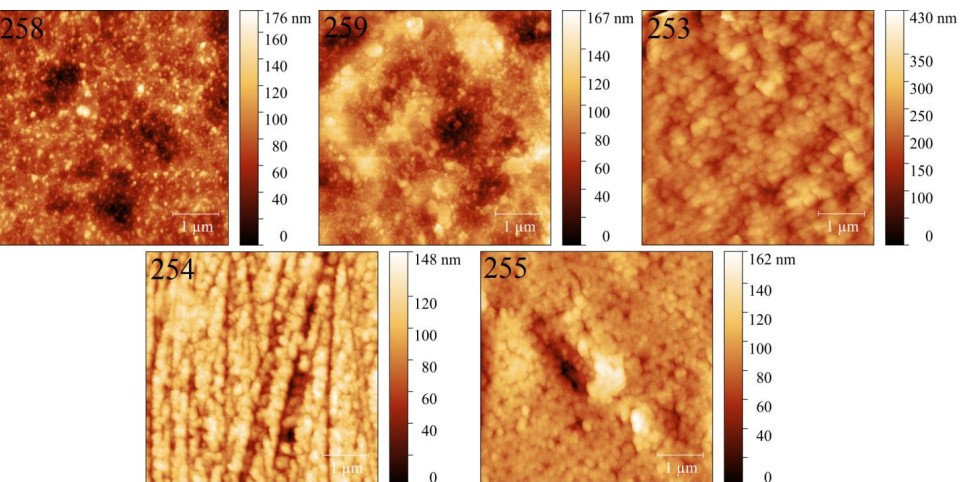

**Figure 2.** AFM images of the por-Si samples with different porosities (arranged according to the increase in the porosity index).

The USXES Si $L_{2,3}$ spectra of porous silicon samples obtained at a depth of analysis equal to 60 nm are presented in Figure 3. Results of the phase analysis simulation for the samples of por-Si with the use of spectra of the reference compounds are presented in Table 2.

The results of the porous silicon compositions with different porosity indexes using USXES technique demonstrated that after the long-term storage of por-Si in the atmosphere, it contains phases of the crystalline, amorphous and disordered silicon as well as the phases of silicon suboxides and dioxide. Samples with low porosity indexes (No. 258, 259) were close compared to the composition of the crystalline silicon; their spectra coincided by more than 95% with the spectrum of the reference c-Si, while the samples with a porosity of more than 30%, a part of oxide phases increased along with porosity index. The percentage of non-oxidized phases relative to the oxidized ones reduced from ~50/50 to 35/65 with an increase in the porosity index at the surface from 30% to 80% (Table 2). This can be explained by a greater surface area of the pores that are subjected to oxidation underexposure in the atmosphere. These data correlate well with the results obtained for porous silicon at the initial stages of its ageing, as well as for micro-, meso- and nanoporous silicon presented in [4,13].

IR transmission spectra of porous silicon samples (Figure 4) include specific features characteristic of the material [14], corresponding to the vibrations of Si–Si (616 cm$^{-1}$), different configurations of Si-Hx (664, 804, 906, 2100–2140 cm$^{-1}$), and Si–O–Si bonds (1060–1170 cm$^{-1}$). Moreover, in the spectra of the samples, there are absorption bands, attributed to $O_2$–Si–OH bonds (~840 cm$^{-1}$), $O_3$–SiH bonds (880 cm$^{-1}$), as well as weakly expressed absorption bands corresponding to the adsorbed $CO_2$ (2360 cm$^{-1}$). A supplement in Figure 4 represents the superposition of the studied spectra presented for a better perception. A comparison of the spectra demonstrates that with an increase in the sample porosity, the degree of the surface oxidation is enhanced. It is revealed as an increase in the intensity of Si–O–Si (1060–1170 cm$^{-1}$) absorption band relative to the other spectral bands, which are in good agreement with the data obtained by the USXES technique. Moreover, the obtained data agree well with the results of the studies concerning the natural ageing of porous silicon presented in [2] and complemented them.

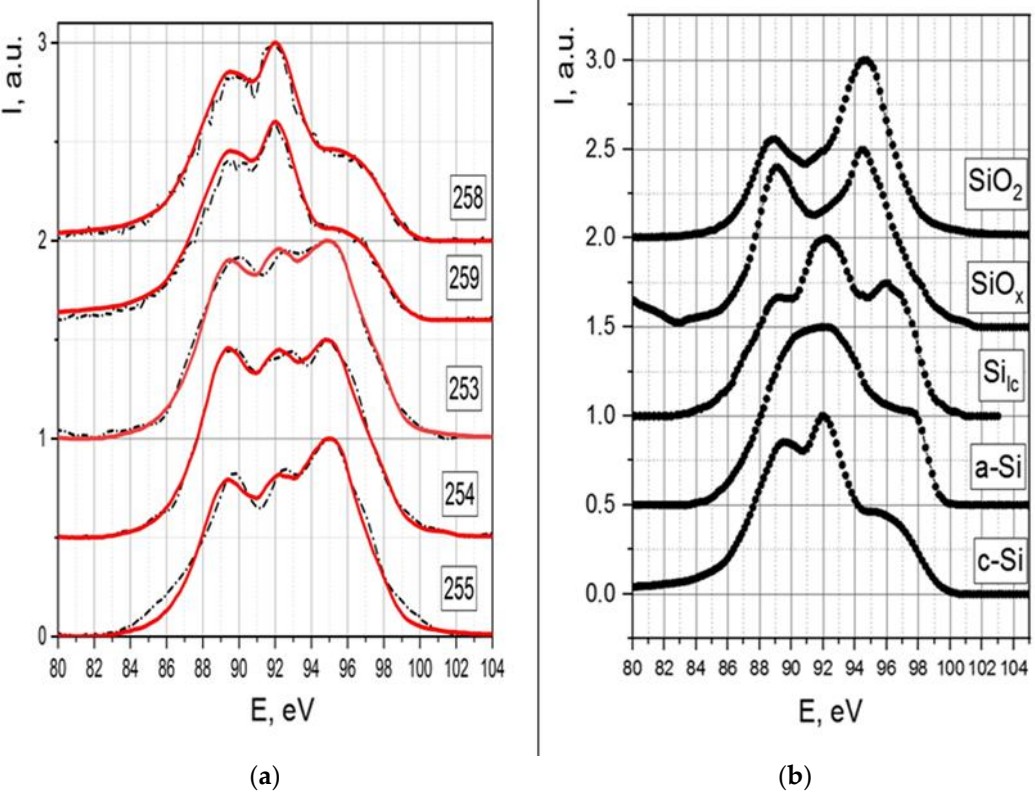

**Figure 3.** USXES Si $L_{2,3}$-spectra of (**a**) porous silicon samples with different porosity and (**b**) spectra of the reference silicon phases. Dashed line—experimental spectrum, solid line—simulated spectrum.

**Table 2.** Distribution of the components of the phase composition as a relative percentage for the samples of porous silicon.

| Depth of Analysis | c-Si, % | a-Si:H/Si$_{lc}$ % | SiO$_x$, % | SiO$_2$, % | Err., % |
|---|---|---|---|---|---|
| No. 253, 60 nm | 34 | 15 | 3 | 48 | 7 |
| No. 254, 60 nm | 36 | 7 | 25 | 32 | 4 |
| No. 255, 60 nm | 29 | 8 | 0 | 62 | 8 |

Photoluminescence spectra of por-Si samples are presented in Figure 5. High-intensive PL peak in the samples with a porosity of more than 30% is arranged in the interval from 600 to 750 nm (Figure 5a). At the same time, por-Si with a porosity of 5 and 14% demonstrated low-intensive PL in the range of 500–700 nm, with PL peak at about 550 nm (Figure 5b). The dependence of the relative PL intensity on the porosity index is presented in Table 1. It should be noted that for the samples with a porosity index below 15%, the intensity and peak position remain almost invariable. A noticeable increase of PL is observed for the porosity index between 15% and 32%. At the same time, an increase in the porosity index within limits from 32% to 80% results in a gradual enhancement of PL peak intensity.

In comparing the luminescence characteristics with the calculated porosity indices, we can conclude that the porous silicon samples with porosity below 30% demonstrate luminescence characteristics for the defect oxide. At the same time, for the samples with porosity of more than 30%, much brighter PL are observed, corresponding to the quantum size effect in the nanocrystals, with diameters of about ~3 nm (within the wavelength range of 650 nm) [15]. A slight shift of the PL band position towards shorter wavelengths can be explained by decreases in the sizes of luminescent nanocrystals with an increase in porosity [2,4]. Comparing the data on photoluminescence and ultra-soft X-ray spectroscopy,

PL within the range of 650–700 nm is more intensive for the samples with less silicon sub-oxide and amorphous silicon at the surface, meaning they can play the role of non-radiative recombination centers in this spectral range.

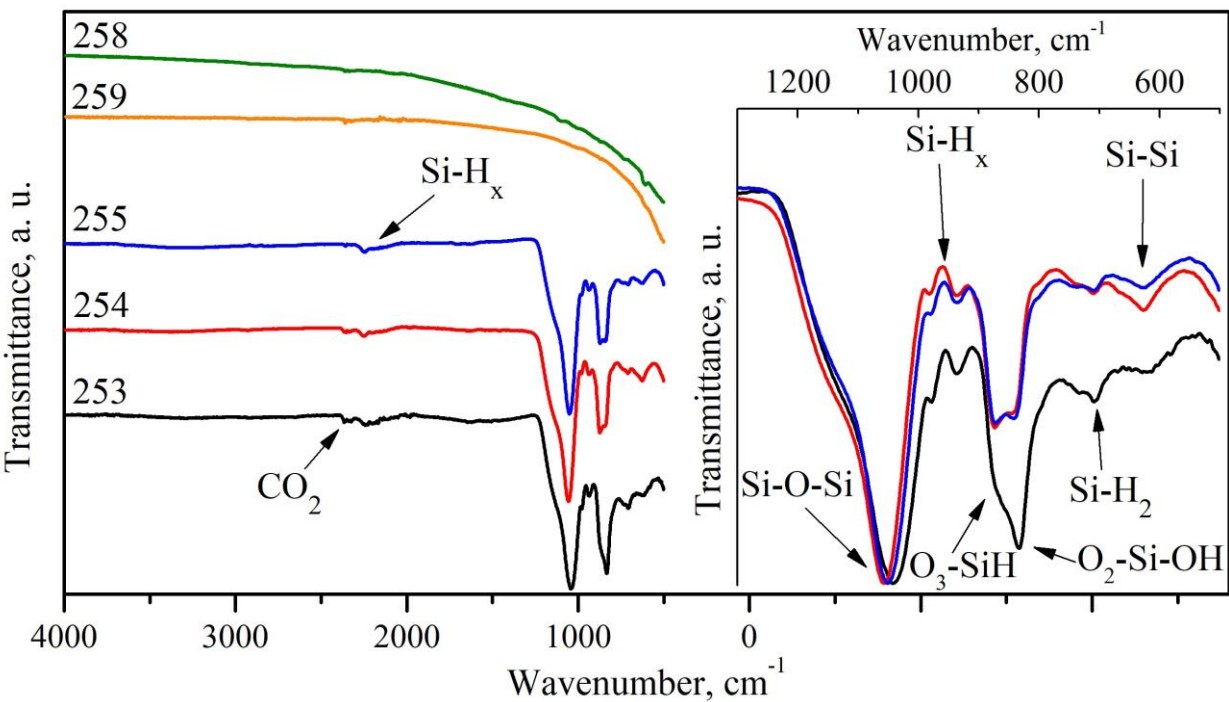

**Figure 4.** IR−spectra of porous silicon samples obtained by ATR technique.

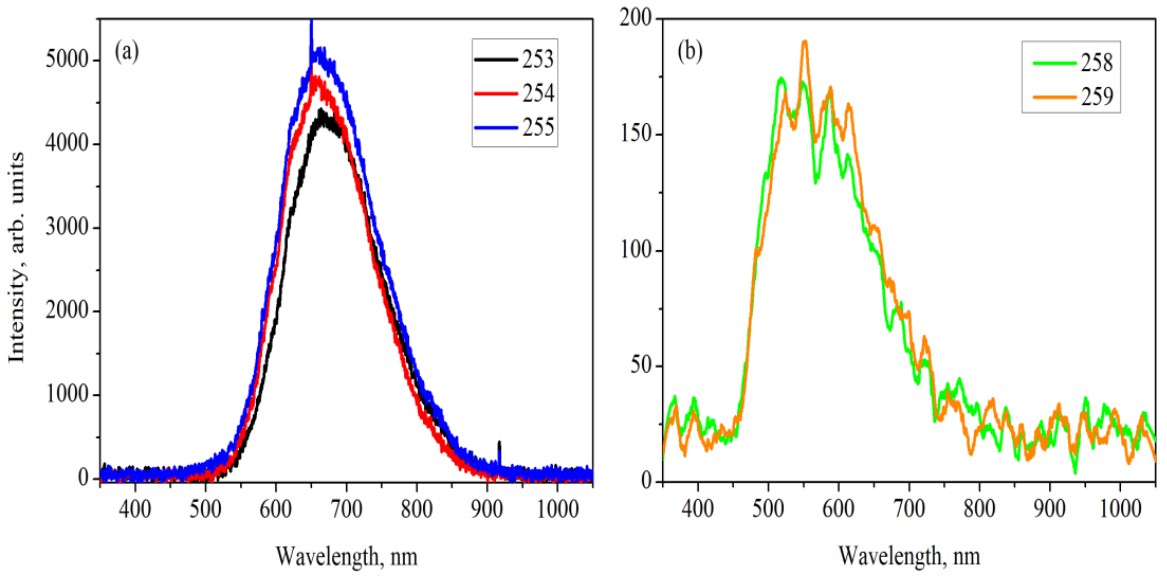

**Figure 5.** Photoluminescence spectra of porous silicon samples (**a**) No. 253–255 and (**b**) No. 258–259.

### 4. Conclusions

Morphology, surface composition and photoluminescence of porous silicon with different porosities were studied in the work. Changes in the PL of the samples can be explained by an increase in silicon porosity when the separation of the sample surface layer into luminescent nanocrystals occurs.

The obtained results demonstrate the efficiency of X-ray reflectometry order in determining the porosity index of porous silicon. It was found that for the samples with low porosity, etching of the silicon surface proceeds without separating the Si plate into nanocrystals and their following oxidation in the atmosphere with a formation of the mixture of the defect oxide $SiO_x$ and $SiO_2$. With an increase in porosity, this separation is realized, thus resulting in the prevailing quantum-size PL mechanism and enhancement of photoluminescence with a peak in the range of 600–750 nm. In addition, the greater the porosity index, the greater the number of silicon nanocrystals formed at the surface layer, thereby enhancing the intensity of photoluminescence. It is considered that porous silicon begins to reveal noticeable photoluminescence properties when its porosity is more than 50% [16]. However, according to our data, porous silicon starts to have luminescent properties when the porosity index is slightly higher than 30%.

**Author Contributions:** Conceptualization, A.S.L.; Methodology, A.S.L.; Investigation, A.S.L., Y.A.P., K.A.B., M.V.G., A.N.L., S.V.K., D.A.M. and O.V.C.; Visualization, O.V.C.; Project administration, A.S.L. All authors have read and agreed to the published version of the manuscript.

**Funding:** This work was funded by grant no. 19-72-10007 from the Russian Science Foundation.

**Institutional Review Board Statement:** The study did not require ethical approval.

**Informed Consent Statement:** Not applicable.

**Data Availability Statement:** Not applicable.

**Conflicts of Interest:** The authors declare no conflict of interest.

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
