# Peer review of "Features of the Composition and Photoluminescent Properties of Porous Silicon Depending on Its Porosity Index"

_coatings, doi:10.3390/coatings13020385_

Round 1
Reviewer 1 Report
In the manuscript, morphology, surface composition and photoluminescence of porous silicon with different porosity were studied. Changes in the photoluminescence of the samples can be explained by the increase of silicon porosity when the separation of the sample surface layer into luminescent nanocrystals takes place. Overall, it was an idea that received my attention and the methodology is technically sound. However, there are some specific issues the authors should address by making modifications before we can proceed and positive action can be taken.
-
The structure of the Introduction section should state the objectives of the work and provide an adequate background, avoiding a detailed literature survey or a summary of the results.
-
When you use an abbreviation in both the abstract and the text, define it in BOTH places upon first use. Further, after you define an abbreviation, use only the abbreviation. Do not alternate between spelling out the term and abbreviating it. i.e., photoluminescence, XRR…
-
The authors show optical properties in Fig. 5. I recommend the authors add a visible spectrum to the figure according to Fig. 5 of [Electronic and optical properties of heterostructures based on transition metal dichalcogenides and graphene-like zinc oxide. Sci. Rep. 2018, 8, 12009, doi:10.1038/s41598-018-30614-3] to make the story clearer and benefit the readers.
-
The English language requires improvements. Spelling and grammatical errors exist in the manuscript. i.e., l. 101: in a quite good agreement; l. 102: look as if they more disordered; l. 138: Inset in the figure 4… We recommend you ask a native English speaker to edit the paper or use an independent professional editor.
- The authors mainly investigate the optical properties of materials, typically based on silicon. Have the authors noticed some similar research on this point? i.e., [Independent degrees of freedom in two-dimensional materials. Phys. Rev. B 2020, 101, 081414(R), doi:10.1103/PhysRevB.101.081414] and [Strain effect on circularly polarized electroluminescence in transition metal dichalcogenides. Phys. Rev. Research 2020, 2, 033340, doi:10.1103/PhysRevResearch.2.033340]… Particularly, [Organocatalytic visible light induced S–S bond formation for oxidative coupling of thiols to disulfides. Tetrahedron 2016, 72, 788–793, doi:10.1016/j.tet.2015.12.036] and [ChemInform Abstract: Organocatalytic Visible Light Induced S—S Bond Formation for Oxidative Coupling of Thiols to Disulfides. ChemInform 2016, 47, doi:10.1002/chin.201623105]
Author Response
1.1. When you use an abbreviation in both the abstract and the text, define it in BOTH places upon first use. Further, after you define an abbreviation, use only the abbreviation. Do not alternate between spelling out the term and abbreviating it. i.e., photoluminescence, XRR…
Our reply:
We agree with the referee's remark and after the abbreviation is determined, only it will be used.
1.2. The authors show optical properties in Fig. 5. I recommend the authors add a visible spectrum to the figure according to Fig. 5 of [Electronic and optical properties of heterostructures based on transition metal dichalcogenides and graphene-like zinc oxide. Sci. Rep. 2018, 8, 12009, doi:10.1038/s41598-018-30614-3] to make the story clearer and benefit the readers.
Our reply: we have no possibility to obtain visible spectra to these samples, therefore we add the reference top the article where the corresponding optical properties were obtained earlier.
1.3. The English language requires improvements. Spelling and grammatical errors exist in the manuscript. i.e., l. 101: in a quite good agreement; l. 102: look as if they more disordered; l. 138: Inset in the figure 4… We recommend you ask a native English speaker to edit the paper or use an independent professional editor.
Our reply: acknowledged and edited
1.3. The authors mainly investigate the optical properties of materials, typically based on silicon. Have the authors noticed some similar research on this point? i.e., [Independent degrees of freedom in two-dimensional materials. Phys. Rev. B 2020, 101, 081414(R), doi:10.1103/PhysRevB.101.081414] and [Strain effect on circularly polarized electroluminescence in transition metal dichalcogenides. Phys. Rev. Research 2020, 2, 033340, doi:10.1103/PhysRevResearch.2.033340]… Particularly, [Organocatalytic visible light induced S–S bond formation for oxidative coupling of thiols to disulfides. Tetrahedron 2016, 72, 788–793, doi:10.1016/j.tet.2015.12.036] and [ChemInform Abstract: Organocatalytic Visible Light Induced S—S Bond Formation for Oxidative Coupling of Thiols to Disulfides. ChemInform 2016, 47, doi:10.1002/chin.201623105]
Our reply: We are mainly engaged with the studies of semiconductors on the basis of silicon and A3B5, and it means some other object domain. Nevertheless, we should note that these articles are useful as a whole and thus they provide the ideas for some new studies.
Reviewer 2 Report
Porous silicon samples with a porosity index of 5 % to 80% were obtained in the work by electrochemical etching and their photoluminescence properties were studied. The more is porosity index the greater is the amount of silicon nanocrystals is formed in the surface layer and, hence, intensity of photoluminescence is enhanced. However, similar conclusion has been widely reported previously. This manuscript is too simple for publication. There is no enough new result. The introduction is too short.
Author Response
2.1. Porous silicon samples with a porosity index of 5 % to 80% were obtained in the work by electrochemical etching and their photoluminescence properties were studied. The more is porosity index the greater is the amount of silicon nanocrystals is formed in the surface layer and, hence, intensity of photoluminescence is enhanced. However, similar conclusion has been widely reported previously. This manuscript is too simple for publication. There is no enough new result. The introduction is too short.
Our reply: The distinction of our work is in the application of X-ray reflectometry and X-ray emission spectroscopy techniques in order to determine porosity and composition of the layer where just photoluminescence is excited. The aim was to set a correct experiment using the above-named mutually-complementary techniques. We have added this idea into the Introduction.
Reviewer 3 Report
In this paper, porous silicon samples are prepared by electrochemical etching, and the porosity index of porous silicon can be controlled between 5% and 80% by the change of the current density during chemical anodic treatment in ECE process and the choice of original single-crystalline plate. The photoluminescence characteristics of the sample were studied in detail. The experimenter calculated the porosity of porous silicon using X-ray reflectometry technique, and studied the photoluminescence characteristics of the sample with the use of fiber-optic spectrometer at a excitation wavelength of 405nm. Through infrared spectroscopy analysis of the sample, it is concluded that with the increase of porosity index, the surface oxide layer of the sample will also increase. Due to the formation of defective oxides, the non-radiative recombination centers increase, and eventually the photoluminescence characteristics of the sample increase with the increase of porosity index. Experimental data show that porous silicon begins to show obvious luminescence properties when the porosity index is greater than 30%. The theoretical and simulation content of this article is very detailed and has a certain reference significance, but the experimental steps need to be further improved. The content of this paper is accurate and clear, the icon is standardized, and it can be properly repaired in some details. It is recommended to consider it after revision.
1.The reference to "all of the critical angles for the samples are below the critical angle of crystalline silicon" suggests adding a critical angle curve of crystalline silicon to Figure 1.
2.The experimental description of the preparation of porous silicon by electrochemical etching is too simple, and it is recommended to add details to the specific experimental steps.
3.The absorption position of CO2(2360 cm-1) is not marked in the infrared spectroscopy of Figure 4. It is recommended to supplement.
4.The format of reference needs to be unified.
5. It is recommended to unify the position and proportion of the illustrations in the article.
Author Response
3.1. The reference to "all of the critical angles for the samples are below the critical angle of crystalline silicon" suggests adding a critical angle curve of crystalline silicon to Figure 1.
Our reply:
According to the referee's suggestion, we added an XRR curve of crystalline silicon to the figure 1.
3.2. The experimental description of the preparation of porous silicon by electrochemical etching is too simple, and it is recommended to add details to the specific experimental steps.
Our reply: Более подробно сделали описание методики получения образцов. Пористый кремний получали анодным электрохимическим травлением пластин кристаллического кремния н (КЭФ) и п-типа (КДБ), при этом ступенчато изменялась плотность тока анодирования. Состав раствор травления при получении образцов не менялся. Все режимы указаны в таблице.
3.3. The absorption position of CO2(2360 cm-1) is not marked in the infrared spectroscopy of Figure 4. It is recommended to supplement.
Our reply:
We have supplemented Figure 4, according to the referee's recommendation.
3.4. The format of reference needs to be unified.
Our reply:
The format of the reference information has been corrected and unified.
Reviewer 4 Report
The authors have presented interesting work on the precisely synthetic and luminescent properties of porous silicon. However, the manuscript should be further improved before its final acceptation for publication. The comments are shown in the attached pdf file. Please make reply point-by-point.

Author Response
How to calculate the porosity values of all samples? According to the formula, two factors shoul be decided: critical angle of porous silicon and substrate silicon. I don't see any available data for calculation?
We added data to Figure 1 .
Please provide detailed meaning or explain these two Si plates (KEF KDB)?
KEF – n-type of crystalline Si, KDP p-type . We added data to article.
Does this treatment mean the samples were performed at different ECE current density with different ECE time? Like 50 mA for 3min, then 20mA for 3min, and 20mA for last 3min?
Exactly
The PL enhancement is evaluated to be more than 20 fold between 15% and 32% porosity. The enhancement should be 'remarkable' instead of 'gradual'.
A noticeable increase of PL is observed for porosity index between 15% and 32%.
Is this average diameter strictly measured? According to the visualization of AFM images, the diameter seemed to be more than 10 nm.
According to the known theory, PL in silicon nanocrystals originates at the sizes of 1-4 nm. Resolution of AFM doesn`t permit to distinct such small objects. Thus we can observe more large objects, providing general information on morphology.
In general, minor inaccuracies have been corrected in accordance with your comments.
Reviewer 5 Report
ABSTRACT
Indicate study design.
Show the values or significance of the results obtained.
INTRODUCTION
Provides well-reasoned and well-founded information on the study.
METHODS
-Include design of study: experimental, observational...
-Indicate the reasons for waiting three months after the samples have been collected to conduct the study.
-In line 80 you say that the investigations were performed after half a year after fabrication of the samples. This is incompatible with the three months mentioned above? It is not clear.
-Include description of statistical analysis performed.
RESULTS AND DISCUSSION
Results and discussion have been presented in the same section. Although the results are very well exposed, there is no explanation for them or a comparison with existing studies. After each result, include an interpretation of the results and/or a comparison with similar backgrounds. You can do it this way or include a separate discussion section of the results.
Author Response
-Indicate the reasons for waiting three months after the samples have been collected to conduct the study.
It is known [4], that for several first days after fabrication of the samples degradation processes in the composition and properties in porous silicon can proceed very rapidly. Therefore, main investigations were performed several months after the samples fabrication when the surface composition and PL properties of the samples are stabilized. We added corresponding explanations to the text of the aerticle.
-In line 80 you say that the investigations were performed after half a year after fabrication of the samples. This is incompatible with the three months mentioned above? It is not clear.
AFM studies were performed 3 months after samples fabrication, all the rest measurements were made half of a year later of the moment of samples fabrication; corrections were made in the text of the article.
-Include description of statistical analysis performed.
Keeping in mind simulation of USXES spectra with the use of the spectra related to the reference phases we provide the reference concerned with this technique [12]$ in order to make it more accessible we added doi (DOI: 10.1016/S0368-2048(00)00393-5) to this article in the reference list.
4.4. RESULTS AND DISCUSSION: Results and discussion have been presented in the same section. Although the results are very well exposed, there is no explanation for them or a comparison with existing studies. After each result, include an interpretation of the results and/or a comparison with similar backgrounds. You can do it this way or include a separate discussion section of the results.
Our reply:
We added the references for comparison and correlations with the previous results.